# Textual Equilibrium Propagation for Deep Compound AI Systems

**Minghui Chen**[1,2], **Wenlong Deng**[2,3], **James Zou**[4], **Han Yu**[1], **Xiaoxiao Li**[2,3*]
[1]Nanyang Technological University   [2]University of British Columbia
[3]Vector Institute   [4]Stanford University

## Abstract

Large language models (LLMs) are increasingly deployed as part of compound AI systems that coordinate multiple modules (e.g., retrievers, tools, verifiers) over long-horizon workflows. Recent approaches that propagate textual feedback globally (e.g., TextGrad (Yuksekgonul et al., 2025)) make it feasible to optimize such pipelines, but we find that performance degrades as system depth grows. In particular, long-horizon agentic workflows exhibit two depth-scaling failure modes: (1) *exploding textual gradient*, where textual feedback grows exponentially with depth, leading to prohibitively long message and amplifies evaluation biases; and (2) *vanishing textual gradient*, where limited long-context ability causes models overemphasize partial feedback and compression of lengthy feedback causes downstream messages to lose specificity gradually as they propagate many hops upstream. To mitigate these issues, we introduce **Textual Equilibrium Propagation (TEP)**, a local learning principle inspired by Equilibrium Propagation in energy-based models (Scellier & Bengio, 2017). TEP includes two phases: 1) a free phase where a local LLM critics iteratively refine prompts until reaching equilibrium (no further improvements are suggested); and 2) a nudged phase which applies proximal prompt edits with bounded modification intensity, using task-level objectives that propagate via forward signaling rather than backward feedback chains. This design supports local prompt optimization followed by controlled adaptation toward global goals without the computational burden and signal degradation of global textual backpropagation. Across long-horizon QA benchmarks and multi-agent tool-use dataset, TEP consistently improves accuracy and efficiency over global propagation methods such as TextGrad. The gains grows with depth, while preserving the practicality of black-box LLM components in deep compound AI system.

## 1 Introduction

Compound artificial intelligence (AI) systems (i.e., pipelines that coordinate large language models (LLMs), retrievers, tools, verifiers, etc.) are becoming a popular way for deploying AI in settings that require multi-step sequential reasoning and external tools (Yao et al., 2023; Shinn et al., 2023). Emerging benchmarks place greater emphasis on long-horizon behaviors such as multi-hop reasoning, iterative planning, and multi-tool interaction at realistic scales (e.g., WebArena (Zhou et al., 2024) for web interaction, and ToolBench (Qin et al., 2024) for large-scale tool calls). As workflow depth and task complexity grow, single-agent architectures reveal fundamental limitations: *they lack the specialized expertise for diverse subtasks (Khattab et al., 2024), accumulate errors without intermediate correction (Shinn et al., 2023)*, and *hit context limits when handling long execution traces (Liu et al., 2024)*.

TextGrad (Yuksekgonul et al., 2025) pioneered "automatic differentiation via text" for compound AI systems. It back-propagates textual feedback from an LLM-as-judge upstream through the computation graph to update agent configurations (e.g., prompts and tools). This approach works well in relatively short chains, steering LLM behaviors without numerical gradients and enabling self-improvement at each step to better handle specialization and error accumulation.

---

*Corresponding author: xiaoxiao.li@ece.ubc.ca

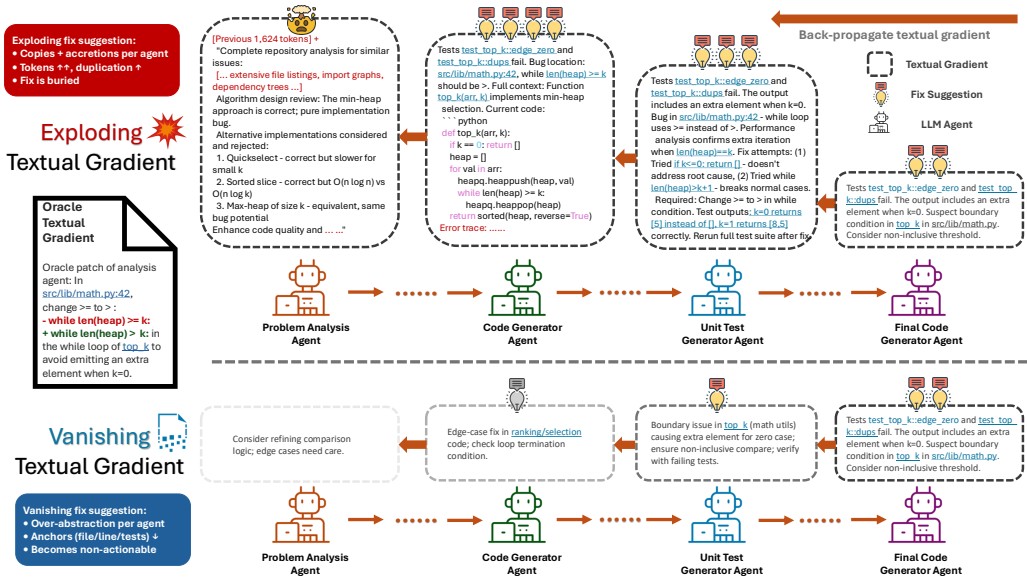

Figure 1: **Textual gradient failure modes in deep compound AI systems**, an example bug report propagates through a multi-agent code generation pipeline (Problem Analyzer → Code Generator → Test Generator → Final Validator). Top–**Exploding gradients:** Each agent adds context to preserve information, causing exponential token growth that buries the actionable one-line fix beneath layers of accumulated texts. Bottom–**Vanishing gradients:** Compression to manage gradient explosion strips away critical specifics (file paths, line numbers, test cases), leaving only generic, non-actionable advice.

However, TextGrad faces depth-scaling challenges in compound AI system analogous to deep neural networks. Like traditional backpropagation's vanishing and exploding gradients (Bengio et al., 1994), textual backpropagation suffers similar instabilities: each layer may amplify noise or weaken useful signals. Specifically, TextGrad exhibits two depth-dependent failures:

- **Exploding textual gradient:** feedback accumulates across layers, leading to growing prompt sizes and LLM-as-judge biases (Zheng et al., 2023). Critical corrections get buried in long contexts, triggering the "lost in the middle" effect (Liu et al., 2024).

- **Vanishing textual gradient:** limited long-context ability causes models to overemphasize early or recent portions (Liu et al., 2024), while compression strategy removes specificity (Chen et al., 2025), leaving early modules with diluted guidance.

Together, these failure modes reveal a fundamental limitation of global textual backpropagation for deep compound AI systems: *optimization signals degrade exponentially as workflow depth increases.*

**Our proposal.** We introduce *Textual Equilibrium Propagation (TEP)*, a local learning framework for compound systems inspired by Equilibrium Propagation in energy-based models (Scellier & Bengio, 2017; 2019). As illustrated in Figure 2, TEP tackles the failure modes of global textual backpropagation using a local, two-phased optimization. In equilibrium propagation, numerical gradients are obtained via two dynamics: a free phase that relaxes to an equilibrium, followed by a nudged phase that gently pushes the system toward task targets. TEP adapts this idea to agentic workflows:

- **Free phase:** local LLM critics iteratively refine prompts at each node in the workflow until reaching an "equilibrium", where no more improvement is suggested by the local critics.

- **Nudged phase:** proximal prompt edits with bounded modification intensity, guided by task-level objectives propagated via forward signaling rather than long backward feedback chains, enabling controlled adaptation toward global goals.

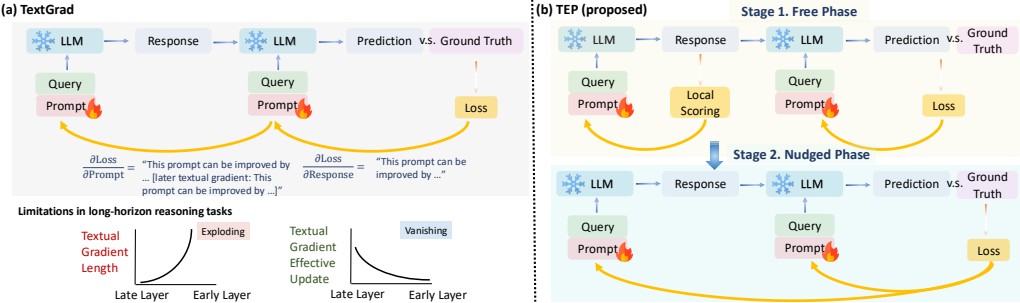

Figure 2: **Overview of Textual Equilibrium Propagation (TEP).** (a) **TextGrad**: Multi-step workflows as Stochastic Computation Graphs (SCGs) where nodes are LLM agents with configurable prompts and edges represent data flow. Global textual backpropagation suffers exploding gradients (exponentially growing feedback) and vanishing gradients (decaying specificity) at depth. (b) **Textual Equilibrium Propagation**: Free phase optimizes each node locally until no further improvements are suggested (equilibrium), then nudged phase applies bounded prompt modifications guided by task objectives. This local approach avoids global feedback chains.

By first achieving local optimization through equilibrium-seeking and then applying controlled proximal edits, TEP reduces both the computational burden and signal degradation associated with global textual backpropagation. As shown in Figure 2, this local approach helps maintain more stable optimization signal as workflow depth increases, effectively mitigating both exploding and vanishing textual gradient issues. TEP also naturally fits with the modular design of modern compound AI frameworks while remaining model-agnostic.

**Contributions.** This paper makes the following three key contributions:

- **Identification of critical gaps:** We analyze fundamental limitations of TextGrad-style optimization in deep compound AI systems, and formalize exploding and vanishing textual gradients as depth-dependent failure modes that make global backpropagation ineffective for long agentic workflow.

- **New optimization method:** We propose Textual Equilibrium Propagation (TEP), a novel compound AI system optimization method specifically designed for deep workflows, with (i) an equilibrium-seeking free phase for local refinement, and (ii) a node-local nudged phase with bounded textual perturbations.

- **Comprehensive empirical validation:** Through extensive experiments on multi-step QA and multi-agent tool-use benchmarks, we demonstrate that TEP consistently outperforms TextGrad (Yuksekgonul et al., 2025) with advantages at greater depths.

## 2 RELATED WORK

**Compound AI System Optimization.** Compound AI systems that orchestrate multiple LLMs, tools, and other components (Wu et al., 2024a; Khattab et al., 2024) have inspired new optimization approaches for multi-step agentic workflows (Yao et al., 2023; Shinn et al., 2023). TextGrad (Yuksekgonul et al., 2025) pioneered automatic differentiation via text, propagating textual feedback through computation graphs to optimize prompts without numerical gradients. While effective in shallow pipelines, TextGrad suffers from depth-dependent issues analogous to exploding and vanishing gradient in deep neural networks (Bengio et al., 1994), where textual feedback grows into intractable long messages. Limited context capacity exacerbates this, as models over-attend to early or recent portions (Liu et al., 2024), and summarization-based compression methods such as those in FedTextGrad (Chen et al., 2025) can dilute feedback into generic suggestions. OPTIMAS (Wu et al., 2025) employs locally trained rewards but requires costly parameter fine-tuning and global evaluations. Exisitng iterative self-refinement approaches (Madaan et al., 2023; Zhang et al., 2025) and prompt optimization methods (Fernando et al., 2024; Dong et al., 2024) still lack a specific de-

sign for deep compound AI systems. TEP mitigates these issues via local equilibrium-based updates that help preserve signal quality even with compound AI system depth increases.

**Long-horizon Reasoning.** Multi-hop QA benchmarks (Yang et al., 2018; Trivedi et al., 2022; Geva et al., 2021) and tool-use datasets (Zhou et al., 2024; Qin et al., 2024) expose fundamental depth-scaling limitations in compound AI systems. Performance tends to drop sharply as compound AI depth increases. This is potentially driven by compounding effects such as the "lost in the middle" problem (Liu et al., 2024), LLM-as-judge biases (Zheng et al., 2023) that distort evaluation, and simple error accumulation across many steps. Self-improvement techniques (Huang et al., 2023; Kumar et al., 2024) provide only partial mitigation, since they fail to decouple local optimization from global feedback chains. Thus, they are still vulnerable to the exponential signal degradation in deep architectures. TEP tackles this limitation through local equilibrium dynamics inspired by energy-based model learning (Scellier & Bengio, 2017; 2019), allowing effective and efficient optimization even at depths where global propagation methods tend to fail.

## 3 THE PROPOSED TEP METHOD

In this section, we formalize deep compound AI systems, define depth-dependent failure modes for global textual feedback, and introduce TEP as a local two-phase learning principle that avoids long-range textual backpropagation. We then show that TEP yields a consistent descent direction and converges under standard conditions.

### 3.1 PROBLEM FORMULATION VIA STOCHASTIC COMPUTATION GRAPHS

We define a compound AI system as a stochastic computation graph (SCG): a directed graph $G = (V, E)$ whose nodes are random variables or deterministic transforms, and whose edges encode functional dependence. Each node $v$ produces an output $z_v$ and has parents $\mathrm{pa}(v) \subseteq V$. We distinguish deterministic nodes $\mathcal{D}$ from stochastic nodes $\mathcal{S}$. In LLM-based compound AI systems, most nodes are stochastic due to the inherent randomness of language model outputs, whereas deterministic nodes typically correspond to simple tools (e.g., calculators, database queries) or function calls with fixed outputs.

**Compound AI system.** Nodes represent LLMs, tools, or controllers with local parameters $\theta_v$ (prompts, hyperparameters, routing rules) and possibly internal state. For deterministic nodes $v \in \mathcal{D}$, we have $z_v = f_v(z_{\mathrm{pa}(v)}; \theta_v)$. For stochastic nodes $v \in \mathcal{S}$, we have $z_v \sim p_v(\cdot \mid z_{\mathrm{pa}(v)}; \theta_v)$. For simplicity, we focus on predefined SCGs and optimize this fixed structure, although the framework and method can naturally extend to dynamic SCGs.

**Problem instance.** Let $D_{\mathrm{task}}$ be a distribution over task inputs $o$. For a fixed SCG $G = (V, E)$ with node-wise parameters $\theta = \{\theta_v\}_{v \in V}$, running $G$ on input $o$ induces a distribution $P_\theta(\cdot \mid o)$ over node outputs $Z = \{z_v\}_{v \in V}$. A task loss $\ell(o, Z)$ then evaluates the resulting output.

**Objective.** We aim to optimize $\theta$ to minimize the expected risk

$$J(\theta) \;=\; \mathbb{E}_{o \sim D_{\mathrm{task}}} \, \mathbb{E}_{Z \sim P_\theta(\cdot|o)} \big[\ell(o, Z)\big]. \tag{1}$$

Equivalently, with $L(o, \theta) := \mathbb{E}_{Z \sim P_\theta(\cdot|o)}[\ell(o, Z)]$, we minimize $J(\theta) = \mathbb{E}_{o \sim D_{\mathrm{task}}}[L(o, \theta)]$.

### 3.2 TEXTUAL GRADIENT PROPAGATION AND DEPTH CHALLENGES

To understand why existing global optimization fails in deep compound AI systems and to motivate our local approach, we first formalize the textual gradient propagation in stochastic computation graphs. We then define exploding and vanishing textual gradients.

**Textual gradient descent in SCGs.** In TextGrad-style optimization, each node $v \in V$ is treated as an LLM agent with local parameters $\theta_v$ (prompts, hyperparameters). A critic LLM evaluates node outputs and generates textual feedback. For a single node, the update is:

$$\theta_v' = U_v(g_v, \theta_v), \quad g_v = C(z_v, \theta_v), \tag{2}$$

where $C$ is the critic LLM that takes node output $z_v$ and current parameters $\theta_v$ as input and generates textual feedback $g_v$ with specific improvement suggestions (e.g., "increase temperature for more

diversity," "add constraint X to the prompt," "revise line Y in the instruction"). The update operator $U_v$ is implemented as an LLM that takes the textual feedback $g_v$ and current parameters $\theta_v$ as inputs, then produces updated parameters $\theta'_v$ by rewriting prompts, adjusting hyperparameters, or generating new instructional text that incorporates the critic's suggestions.

For multi-node chains, the update at node $v$ depends on downstream feedback:

$$\theta'_v = U_v(g_v, \theta_v), \quad g_v = C(z_v, \theta_v, g'), \tag{3}$$

where $g'$ aggregates textual gradients from descendant nodes. This coupling creates a dependency chain: updates at depth $d$ require propagating feedback through all subsequent layers, leading to the depth-scaling failure modes we formalize below.

**Exploding Textual Gradient.** While exploding numerical gradient manifests as unbounded magnitudes, exploding textual gradient show up as unbounded message length and complexity. Let $B(g)$ denote the token count required to transmit feedback $g$ (a practical constraint that TEP's bounded local updates respect). Explosion occurs when maintaining useful feedback (specificity $S(g_u) \geq \tau$ for some threshold $\tau > 0$) across a length-$k$ chain requires exponentially growing messages:

$$\mathbb{E}[B(g_u)] \geq c\gamma^k \quad \text{for } \gamma > 1, \ c > 0. \tag{4}$$

The key distinction from numerical gradients is that backpropagated textual feedback grows exponentially with SCG depth, potentially exceeding LLM context limits (e.g., 128K tokens). Each layer must preserve all downstream feedback while adding its own analysis, so the gradient message becomes not only very long but also increasingly distorted as LLMs inject biases that compound through the chain. As a result, exploding textual gradient can cause transmitted message to exceed an LLM's context window or hide essential information inside an overly long context.

**Vanishing Textual Gradient.** Analogous to numerical gradients that decay exponentially in deep networks, textual gradients can vanish when attempts to compress exploding feedback cause specificity to degrade over multi-hop propagation. Let $S(g) \in [0, 1]$ measure the actionable specificity of feedback $g$ (how well it guides parameter updates). For a length-$k$ chain from the final output back to node $u$, vanishing occurs when:

$$\mathbb{E}\big[S(g_u)\big] \leq C\alpha^k \quad \text{for } \alpha \in (0, 1), \ C > 0, \tag{5}$$

where $g_u$ is the feedback reaching $u$. Unlike numerical gradients that shrink in magnitude, textual gradients lose information density as compression and summarization attempt to manage the exponentially growing context. This is related to the "lost in the middle" effect, where LLMs struggle to extract information buried in long contexts (Liu et al., 2024). For instance, actionable feedback like "replace >= with > in the loop condition" may degrade into generic suggestions such as "improve algorithm efficiency" after several attempts to compress the message. In this paper, we quantify vanishing textual gradients through a low effective update rate, meaning that suggested prompt updates do not lead to performance improvements.

**Implications for compound AI systems.** These failure modes limit TextGrad scalability in deep SCGs. Exploding gradients lead to context overflow and high computational costs, while vanishing gradients leave upstream nodes with generic, unusable feedback. In practice, even modest depths (10 to 20 nodes) can trigger these issues, making global textual backpropagation impractical for real systems.

This motivates our local approach. TEP uses node-local critics (avoiding $g'$ dependencies) and bounded parameter updates. As shown in Appendix A, local evaluation can maintain feedback quality $S(g) \geq \tau$ within context limits $B(g) \ll$ context limit, circumventing both explosion and vanishing while keeping the effectiveness of textual optimization.

### 3.3 OUR PROPOSED SOLUTION: TEXTUAL EQUILIBRIUM PROPAGATION (TEP)

To address vanishing and exploding textual gradients, we introduce *Textual Equilibrium Propagation (TEP)* for optimizing compound AI systems using textual gradients. TEP adapts equilibrium propagation from numerical networks to textual SCGs, replacing global critique propagation with local, bounded perturbations around a node-local equilibrium. Figure 2 (b) shows an overview of the proposed TEP, which consists of two main phases: a free phase and a nudge phase. Note that prompt optimization tasks we evaluated only provide global label at the final output level. No intermediate

ground-truth labels are available. By default, the used critic model is the same as node model. These phases are detailed below.

**Free phase.** In TEP, each node $v$ first optimizes locally to reach a behavioral equilibrium using two learnable mechanisms: (i) an LLM critic with a structured rubrics prompt $\theta_v^{\text{critic}}$ that evaluates outputs using both task-independent quality metrics (clarity, completeness, consistency) and task-dependent performance criteria, and (ii) an actor temperature parameter $\theta_v^{\text{temp}} \sim \mathcal{U}(0.3, 0.9)$ that controls exploration-exploitation during local refinement. The critic scores output $z_v$ and returns feedback $g_v = C(z_v, \theta_v^{\text{critic}})$, where $C$ is the critic defined in Section 3.2, now applied locally without descendant gradient $g'$. Free phase reaches equilibrium $x_\star^0(\theta)$ when the critic's scores stabilize across iterations. This local refinement removes global feedback chains with the structured critic prompt. (See Appendix C for implementation details.)

**Nudged phase.** In classical equilibrium propagation, a small perturbation is applied at the outputs to steer the system toward the desired behavior. In TEP, the perturbation is implemented as a *minimal prompt edit* at each node in a SCG of deep compound AI system. Concretely, these edits are applied to reinforce node-local criteria that align with the global task target. After applying the edits, the system is run again and iterate until it reaches a *nudged equilibrium*, which is different from free equilibrium based on the local feedback signal.

**Local update rule.** Unlike classical EP that forms updates from numerical gradient differences between free and nudged equilibria, TEP adopts an LLM-defined operator to jointly combine feedback from both phases. The update follows: $\theta_v' = U_v(g_v^f, g_v^n, \theta_v)$, where $g_v^f$ and $g_v^n$ are the feedback signals from the free and nudged phases, respectively. Both are kept within bounded length and specificity in Section 3.2's definitions. $U_v$ is the LLM-defined update operator. It maps these feedback signals to an updated node specification (i.e., prompt edit). We apply validation-based selection at every update (in both phases and in the final combination), keeping only edits that do not reduce validation performance.

## 4 EXPERIMENTAL EVALUATION

We evaluate TEP on diverse compound AI tasks that require long-horizon reasoning, and compare it against state-of-the-art textual optimization methods. Our experiments are designed round three key questions: (1) Does TEP improve performance over global textual backpropagation methods? (2) How severe are exploding/vanishing gradient issues in practice, and does TEP mitigate them? (3) Which components of TEP contribute most to its overall effectiveness?

### 4.1 EXPERIMENT SETUP

**Tasks and Datasets.** We evaluate on four multi-step reasoning and tool-orchestration benchmarks:

1. **PubMedQA** (Jin et al., 2019) (biomedical QA). We use the original "expert" split and discard ambiguous samples, yielding 475/25/500 (train/dev/test) question–abstract pairs. Metric: classification accuracy over {yes,no,maybe}. Base model: `GPT-4o`. Prompt-optimization modules: *Context Analyst*, *Problem Solver*.

2. **STARK-PRIME** (Wu et al., 2024b) (retrieval over semi-structured biomedical corpora). We use the original split 495/51/96 queries. Metric: Mean Reciprocal Rank (MRR). Base model: `Claude 3 Haiku`. Prompt-optimization modules: *Text Scorer*, *Relation Scorer*.

3. **HotpotQA** (Yang et al., 2018) (multi-hop QA with retrieval-augmented generation). We keep the official splits 1000/250/100. Metric: answer-level F1. Base model: `GPT-4o-mini`. Prompt-optimization modules: *Question Rewriter*, *Info Extractor*, *Hint Generator*, *Answer Generator*.

4. **BigCodeBench** (Zhuo et al., 2024) (self-verified code generation). We proportionally subsample to 500/25/70 coding tasks. Each item includes a natural-language spec and reference unit tests. Metric: *pass@1*. Base model: `Claude 3 Haiku`. Prompt-optimization modules: *Code Generator*, *Unit-Test Generator*, *Final Code Generator*.

**Baselines.** We compare TEP against five strong baselines, summarized below.

1. **Chain-of-Thought (CoT)** (Wei et al., 2022): Standard prompting with step-by-step reasoning; no learning or prompt updates. Serves as the unoptimized reference.

2. **DSPy** (Khattab et al., 2024): A modular programmatic framework that learns and compiles prompts from demonstrations. We use task-specific modules with default DSPy tuning recipes.

3. **Hierarchical Behavior Cloning (HBC)** (Le et al., 2018): Hierarchical imitation learning that aligns sub-module outputs to high-reward trajectories.

4. **TextGrad** (Yuksekgonul et al., 2025): Global textual backpropagation over the full SCG. Feedback is propagated hop-by-hop without length constraints, following the original setup.

5. **TextGrad+Summarization** (Chen et al., 2025): We adapt summarization prompting—originally used in cross-device federated learning to compress stacked-layer textual gradients and to control their length.

**Implementation Details.** Across all benchmarks, we refer the SCG configurations of Wu et al. (2025). We do not include a direct comparison to OPTIMAS, as their focus on parallel, parameter fine-tuning differs from our black-box prompt-optimization setting. By default, TEP uses *20* iterations in the free phase and *40* iterations in the nudged phase. We detail the computational cost of each baselines in the Appendix D.4.

## 4.2 MAIN RESULTS

Table 1: Performance comparison across compound AI benchmarks. Best in **bold**.

| Method | PubMedQA Medical Analysis (Acc.) | STARK-PRIME Complex Retrieval (MRR) | HotpotQA RAG (F1) | BigCodeBench Verified Code Gen. (Pass Rate) |
|---|---|---|---|---|
| CoT | $57.34\pm1.12$ | $39.76\pm0.84$ | $33.92\pm0.76$ | $34.15\pm1.43$ |
| HBC | $58.80\pm0.58$ | $36.95\pm0.59$ | $21.16\pm0.97$ | $27.78\pm2.08$ |
| DSPy | $60.26\pm0.40$ | $41.40\pm0.04$ | $44.90\pm0.32$ | $33.81\pm2.75$ |
| TextGrad | $56.96\pm2.24$ | $41.31\pm1.67$ | $24.86\pm1.19$ | $35.71\pm0.10$ |
| TextGrad w/ Sum | $56.12\pm1.85$ | $40.72\pm1.21$ | $24.12\pm1.25$ | $35.12\pm0.67$ |
| TEP (Ours) | $\mathbf{62.02\pm1.31}$ | $\mathbf{42.72\pm0.65}$ | $\mathbf{48.72\pm1.32}$ | $\mathbf{38.97\pm0.39}$ |

**Overall Performance Result.** Table 1 shows that TEP perform consistently well across diverse compound AI benchmarks. It achieves the best performance on all four tasks, with particularly strong gains on complex reasoning benchmarks: $8.1\%$ improvement over the next-best method on HotpotQA and $3.4\%$ on BigCodeBench. TextGrad with Summarization shows modest improvements over vanilla TextGrad on retrieval tasks (STARK-PRIME) but its performance drops on reasoning-heavy tasks (HotpotQA, BigCodeBench), suggesting that compression-based solutions cannot fully address textual gradient issues. The results validate that TEP's local optimization approach scales effectively to real-world compound AI systems while avoiding the fundamental limitations of global textual backpropagation.

## 4.3 EMPIRICAL ANALYSIS OF TEXTUAL GRADIENT FAILURE MODES

We empirically validate our theoretical analysis with controlled depth-scaling experiments that isolate the effects of workflow complexity on textual gradient propagation. Using artificially deepened compound AI systems, we demonstrate the practical manifestation of exploding and vanishing gradient phenomena predicted by our hypothesis.

**Experimental Design.** We vary computational graph depth using the BigCodeBench code generation pipeline as our base architecture. The standard pipeline contains four sequential modules: Problem Analysis, Code Generation, Test Generation, and Code Refinement. To study depth effects, we introduce a *scale factor* $s \in \{1, 2, 3, 4, 5\}$ that replicates each module with intermediate refinement stages. At scale $s$, the graph $\mathcal{G}_s$ contains $4s$ nodes performing semantically equivalent transformations—code reformatting, documentation enhancement, and style consistency checks—that

preserve functional correctness while extending gradient propagation paths. We measure two key metrics: *feedback token count $B(s)$* quantifying message complexity, and *effective update rate $\rho(s)$* measuring the fraction of nodes that successfully improve their outputs based on received feedback.

**Exploding Textual Gradients: Exponential Message Growth.** Figure 3(a) shows that TextGrad's feedback messages grow roughly exponentially with depth, and the accumulated text length can exceed an LLM's maximum context window as the depth increases. The token count increases from $2K$ at $s = 1$ to over $32K$ at $s = 5$, approaching the context limits of modern LLMs. This exponential scaling ($B(s) \sim 2.2^s$) matches our hypothetical prediction: to preserve actionable information across multiple hops, messages need to become exponentially longer.

**Vanishing Textual Gradients: Information Decay Under Compression.** When TextGrad uses summarization to manage token budgets (TextGrad w/ Sum), gradient specificity decays quickly with depth. Figure 3(b) shows effective update rates declining from $36\%$ at $s = 1$ to merely $5\%$ at $s = 5$. Compressed feedback loses actionable details—transforming specific corrections like "fix undefined variable `result` on ..." into generic suggestions like "improve code quality"—validating our analysis of information decay under bounded token constraints.

**TEP's Graceful Degradation.** TEP exhibits fundamentally different scaling behavior, keeping nearly constant token complexity and showing only modest performance degradation ($37\%$ to $33\%$ update success rate). By eliminating multi-hop gradient chains and relying on local equilibrium computations, TEP avoids the exponential failure modes while achieving consistent optimization effectiveness across different scaling levels of workflow.

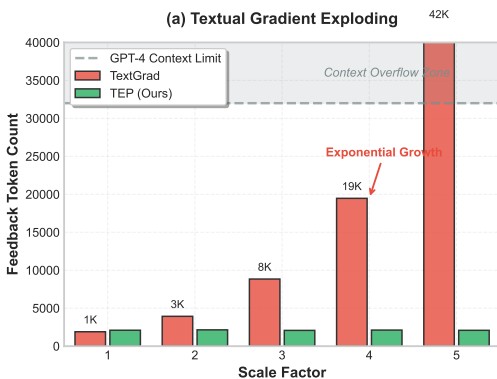 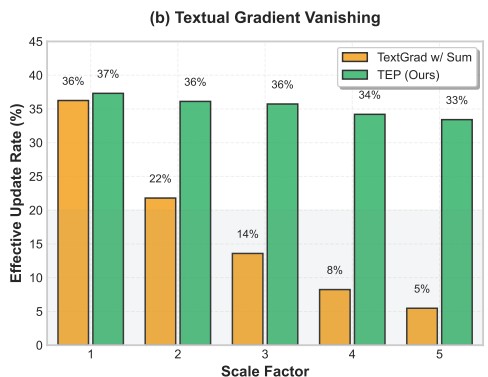

Figure 3: Textual gradient failure modes with increasing workflow depth on BigCodeBench. (a) **Exploding gradients**: TextGrad feedback messages grow exponentially, reaching context overflow at scale factor 5. (b) **Vanishing gradients**: Compressed feedback (TextGrad w/ Sum) loses specificity, leading to sharp drop in effective update rates. TEP maintains stable performance across all scales through local optimization.

## 4.4 SOLUTION OPTIMIZATION EXPERIMENTS

**Solution-Optimization Setting.** We evaluate TEP on *solution optimization*, where the target is to improve the model's *reasoning outputs* rather than its input prompts. We use two benchmarks: (i) **GPQA** (Rein et al., 2024)—expert-level physics/biology/chemistry questions requiring multi-step reasoning (following Yuksekgonul et al., 2025); and (ii) an adapted **Object Counting** dataset with 5-step hierarchical arithmetic (e.g.,"4 pallets, each with 3 crates, each with 5 boxes, each with 2 bags of 40 screws; 360 discarded—how many remain?").

Table 2: Solution optimization performance.

| Method | GPQA (Acc.) | Object Counting (Acc.) |
|---|---|---|
| CoT | 38.5 | 72.4 |
| TextGrad | 41.0 | 74.2 |
| TextGrad w/ Sum | 39.5 | 68.9 |
| TEP (Ours) | **44.5** | **81.6** |

This setting tests whether TEP's local, task-independent critic and proximal prompt edits remain beneficial when optimization acts on generated *solutions*. For GPQA, optimization targets a single node. This does not reduce TEP to TextGrad: TEP's task-independent critic and proximal prompt edits remain effective even in single-node prompt optimization. For Object Counting, intermediate, verifiable subresults make the task well-suited to TEP's free-phase local refinement. In contrast, global textual backpropagation (TextGrad) continues to suffer depth-related exploding/vanishing feedback when multi-step reasoning induces long reformulation chains (Appendix E).

**Setup.** We compare four methods: **CoT** (baseline), **TextGrad** (full backpropagation), **TextGrad+Summ.** (100-token feedback cap), and **TEP** (local refinement). Models: `GPT-4` on 200 GPQA questions and `Llama-3.2` on 500 Object Counting problems.

**Results.** Table 3 shows TEP outperforms all baselines. On GPQA, TEP reaches $44.5\%$ accuracy, a 3.5 point gain over TextGrad; TextGrad+Summ. falls below CoT ($39.5\%$), indicating that compression weakens nuanced reasoning signals. On Object Counting, TEP attains $81.6\%$vs. TextGrad's $74.2\%$, while summarization leads to a larger drop ($68.9\%$). These results highlight TEP's robustness for solution optimization, where precise, depth-robust feedback is critical.

### 4.5 ABLATION STUDIES

**Ablation on TEP Components.** Figure 4 studies the contribution of TEP's two phases on HOTPOTQA and BIGCODEBENCH by removing one component at a time while holding all other settings (models, iterations, hyperparameters) fixed.

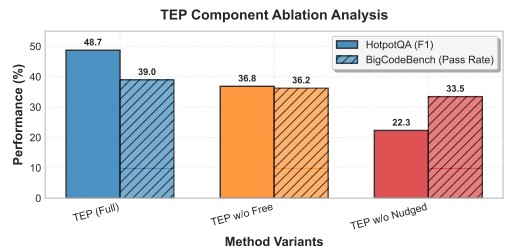

*Removing the nudged phase* leads to severe degradation—on HOTPOTQA the score collapses to $22.3\%$ F1 (a $26.4$ point drop). This shows that purely local equilibrium (free phase) is not enough to achieve system-level coordination: nodes converge to isolated local optima that fail to compose into globally coherent solutions. A similarly severe decline is observed

Figure 4: Component ablation analysis demonstrating the necessity of both TEP phases across task types.

qualitatively on BIGCODEBENCH, underscoring the importance of nudged, task-aligned coordination.

*Removing the free phase* also hurts performance, though less dramatically: $-11.9$ F1 on HOTPOTQA and $-2.7$ pass@1 on BIGCODEBENCH. The free phase provides high-quality local optima that serve as stable starting points for effective nudged updates. Without this local refinement, nudging operates from weaker initializations and its coordinating effect is reduced. The smaller impact on BIGCODEBENCH suggests code generation is more tolerant to imperfect local optima than multi-hop reasoning chains, where errors compound more easily.

**Takeaway.** The results support TEP's *synergistic* design: the free phase establishes strong local configurations; the nudged phase aligns them toward global objectives. Neither local refinement nor global coordination alone suffices. Both are necessary to achieve TEP's full gains.

## 5 CONCLUSIONS

In this paper, we introduced Textual Equilibrium Propagation (TEP), a local learning principle that tackles the depth-scaling limitations of existing textual optimization methods in compound AI systems. By formalizing exploding and vanishing textual gradients as core challenges in deep agentic workflows, we showed how global backpropagation becomes impractical as system depth increases. TEP adapts equilibrium propagation to textual SCGs with a two-phase approach (free and nudged phases) that supports bounded, node-local updates while still leveraging LLMs' natural reasoning capabilities for compound AI system optimization. Our empirical evaluation demonstrates that TEP consistently outperforms TextGrad and other iterative self-refinement approaches across multi-step

QA and multi-agent tool-use benchmarks, with gains that become larger at greater depths. The local, equilibrium-based approach shows a path toward optimizing deep compound AI systems. In subsequent research, we plan to explore dynamic graph optimization and large-scale multi-agent coordination.

## 6 ACKNOWLEDGMENT

The research is supported, in part, by the NSERC Discovery Grant RGPIN-2022-05316, NSERC Alliance Grant ALLRP 602633-24, Tri-Agency Canada; Canada CIFAR AI Chair Awards, and Canada Research Chair Fellowship; IITP grant, the Ministry of Science and ICT (RS-2024-00445087, RS-2025-25464461), funded by the Korea government (MSIT); the Ministry of Education, Singapore, under its Academic Research Fund Tier 1 (RG101/24).

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

## A   ANALYSIS OF TEXTUAL GRADIENT FAILURE MODES

This section analyzes why global textual backpropagation fails at depth and how TEP avoids these issues.

### A.1   SETUP AND ASSUMPTIONS

**Textual feedback propagation.**   In an SCG with node outputs $\{z_v\}$ and parameters $\{\theta_v\}$, global textual signals from the final output $F$ pass through intermediate LLMs, creating messages $G^{(1)}, G^{(2)}, \ldots, G^{(k)}$ before reaching upstream nodes.

**Information contraction.**   Each reformulation loses task-relevant information. We model this with a per-hop factor $\alpha \in (0, 1)$ where mutual information about desired improvements $\Delta Q$ decays geometrically across hops.

**Bounded channel capacity.**   A textual message with $B$ tokens carries at most $\kappa B$ bits of task-relevant information. Initial information is budget-limited, and reformulations can only maintain or reduce it.

### A.2   THE DEPTH TAX

**Exponential decay.**   Combining contraction with bounded capacity yields

$$I(\Delta Q; G^{(k)}) \ \leq \ \kappa B \, \alpha^k.$$

To maintain signal quality at distance $k$, message length must grow as $B \geq \alpha^{-k}$ (explosion); under fixed budgets, signals decay as $\alpha^k$ (vanishing).

**The fundamental trade-off.**   Global textual backpropagation faces exponentially growing context requirements or exponentially decaying signal quality.

### A.3   HOW TEP AVOIDS DEPTH ISSUES

**Local contrasts replace chains.**   TEP uses local two-phase contrasts (free vs. nudged) at each node. Critics evaluate node outputs directly, producing $\mathcal{O}(1)$-sized signals whose quality depends on local context, not graph depth.

**Depth-invariant updates.**   Under mild conditions on critic informativeness, each node maintains consistent update signal quality. TEP's optimization quality scales with node count, not depth.

### A.4   CONVERGENCE PROPERTIES

**Local equilibrium.**   When a node's refinement operator $T_v$ is contractive (rate $\gamma < 1$), free-phase equilibrium exists, is unique, and converges geometrically at rate $\gamma$—independent of graph depth.

**Complexity comparison.** Global methods face per-iteration costs $\rho^d$ with decaying signal quality $\alpha^d$. TEP maintains constant per-node context and avoids multi-hop decay, achieving $\mathcal{O}(1)$ per-node complexity with natural parallelization.

**LLM-as-Judge bias and TextGrad.** Textual gradient methods treat LLM feedback as a kind of "gradient" for optimizing a compound AI system. This relies heavily on the LLM acting as a judge, so feedback quality is crucial for stable prompt optimization and convergence. In practice, the critic (or judge) model needs to be strong in the target domain, which is why we prefer frontier LLMs for the more complex prompt-optimization tasks. In deep compound AI systems, bias can accumulate. A small judgement error in a downstream node can compound as it is backpropagated upstream. TEP addresses this specifically through the Free Phase and Nudged Phase. In the free phase, iterative local updates under a rubric-based critic stabilize each node. In the nudged phase, proximal prompt edits are guided directly by the final task objective (i.e., final ground truth labels). Together, these steps reduce the avoid the systematic bias (as shown in section 4.3) that accumulates when feedback is propagated step by step along the chain.

**Model Choice and Model Size.** In experiments, our choice of LLMs is driven by task difficulty and domain: for complex prompt-optimization tasks requiring long-horizon reasoning and reliable self-critique we use frontier models such as GPT-4o and Claude, while for simpler solution-optimization tasks we use lighter models such as Llama 3, which already achieve strong base performance and are more cost-efficient. As in TextGrad, TEP assumes a base LLM that is capable of meaningful self-correction on the target task: very small models or models lacking domain knowledge often fail to generate or evaluate high-quality refinements, which limits the effectiveness of any textual optimization framework.

**Summary.** Global textual feedback incurs exponential depth costs through information loss and context growth. TEP generates signals locally, maintaining stable quality and depth-robust convergence.

## B  TEP ALGORITHM PSEUDOCODE

This section provides a compact pseudocode for Textual Equilibrium Propagation (TEP), highlighting the two-phase optimization and node-local updates.

---

**Algorithm 1: Textual Equilibrium Propagation (TEP)**

```
1  Input:  SCG G=(V,E); task distribution D_task; beta>0; epsilon>0; T_max
2  Output: Parameters theta={theta_v}_v
3
4  // Initialization
5  for each v in V:
6    theta_v^critic <- structured critic prompt (App. D)
7    theta_v^temp   ~  U(0.3, 0.9)
8    theta_v^actor  <- initial actor prompt
9
10  for t = 1..T_max:
11    o ~ D_task
12
13    // Phase 1: Free-phase equilibria (parallel over v)
14    parallel for v in V:
15      repeat
16        z_v <- LLM(theta_v^actor, z_pa(v); temp=theta_v^temp)
17        g_v^free <- Critic(z_v, theta_v^critic)
18        score_v  <- extract_score(g_v^free)
19      until score_v stabilizes
20      x_v^0 <- z_v
21
22    // Phase 2: Nudged-phase equilibria (parallel over v)
23    parallel for v in V:
```

```
24      ell_v    <- local_objective(x_v^0, o)
25      nudge_v <- generate_nudge(ell_v, beta)
26      theta_v^nudged <- theta_v^actor + nudge_v
27      repeat
28        z_v <- LLM(theta_v^nudged, z_pa(v); temp=theta_v^temp)
29        g_v^nudged <- Critic(z_v, theta_v^critic)
30        score_v    <- extract_score(g_v^nudged)
31      until score_v stabilizes
32      x_v^beta <- z_v
33
34    // Phase 3: Local update (parallel over v)
35    parallel for v in V:
36      theta_v^new <- UpdateOperator(g_v^free, g_v^nudged, theta_v^actor)
37      theta_v^actor <- theta_v^new
38      if performance improved then
39        theta_v^temp <- max(0.3, theta_v^temp*0.95)
40      else
41        theta_v^temp <- min(0.9, theta_v^temp*1.05)
42
43    if |J(theta^t) - J(theta^{t-1})| < epsilon then break
44    beta <- beta * 0.9
45
46  return {theta_v^actor, theta_v^critic, theta_v^temp}_v
```

**Key algorithmic properties.** The pseudo code illustrates several critical aspects of TEP: (1) **Local parallelism**: Each node $v$ independently seeks equilibrium without requiring global coordination, enabling parallel execution across all nodes. (2) **Two-phase structure**: The free phase establishes high-quality local optima through critic-guided refinement, while the nudged phase coordinates these solutions toward global objectives through minimal textual perturbations. (3) **Bounded complexity**: Each node processes only local feedback signals $(g_v^f, g_v^n)$ of constant size $\mathcal{O}(1)$, avoiding the exponential message growth that plagues global textual backpropagation. (4) **Adaptive exploration**: Temperature parameters $\theta_v^{\text{temp}}$ automatically adjust based on local performance, balancing exploration and exploitation at each node. (5) **Parameter annealing**: The nudging strength $\beta$ decreases over iterations, allowing strong initial coordination that gradually transitions to fine-tuning as the system converges.

## C   FREE PHASE IMPLEMENTATION DETAILS

This section provides concrete implementation details for TEP's free phase, including the structured critic prompt design and temperature tuning strategy. Note that the descriptions and examples are meant to illustrate the prompt-design principles. In practice, we further refine each task prompt using frontier LLMs to better match task requirements.

### C.1   STRUCTURED CRITIC PROMPT AND RUBRIC SYSTEM

TEP employs a two-component rubric system that separates task-independent quality metrics from task-dependent performance criteria. This separation enables consistent quality assessment while allowing task-specific adaptation.

**Task-independent quality metrics.** These metrics remain fixed across all nodes and evaluate general output quality through six dimensions:

- **Structural Clarity (1–5):** Output organization and unambiguity.
  - 1: Disorganized, ambiguous structure
  - 3: Basic structure present but lacks coherence
  - 5: Well-organized with clear logical flow and explicit sections
- **Specification Completeness (1–5):** Coverage of input requirements.

- 1: Major requirements missing
- 3: Core requirements addressed, details lacking
- 5: All requirements fully addressed with appropriate depth

- **Internal Consistency (1–5):** Freedom from contradictions.
  - 1: Multiple contradictions present
  - 3: Minor inconsistencies in edge cases
  - 5: Fully consistent across all aspects

- **Context Integration (1–5):** Utilization of parent node outputs.
  - 1: Ignores available context
  - 3: Uses context but misses connections
  - 5: Seamlessly integrates all relevant context

- **Reasoning Transparency (1–5):** Clarity of decision rationale.
  - 1: No justification for choices
  - 3: Some reasoning provided but gaps remain
  - 5: Clear rationale for all significant decisions

- **Format Compliance (1–5):** Adherence to expected output schema.
  - 1: Free-form, ignores format requirements
  - 3: Partial compliance with format
  - 5: Strict adherence to specified schema/structure

## D  TEP LOCAL CRITIC EVALUATION TEMPLATES

This appendix provides concrete examples of TEP's local critic evaluation process. The following displays illustrate the free phase context, nudged phase context, and local critic prompt using the multi-agent code generation pipeline scenario from Figure 1. These examples demonstrate how TEP's local optimization approach handles the specific debugging task where textual gradients would otherwise explode or vanish through the Problem Analyzer $\rightarrow$ Code Generator $\rightarrow$ Test Generator $\rightarrow$ Final Validator chain.

---

**Free Phase Context (Node $v$)**

**System:** You are a local verifier for node $v$. Evaluate outputs against criteria below.

**Node Inputs (abridged):**

```
query: "Explain the failure
  case for k=0"
retrieved: ["src/lib/math.py:
  top_k"]
```

**Free Output:**

```
The result length is off by one
  for k=0 ...
```

---

**Nudged Phase Context (Node $v$)**

**Nudge:** Encourage stricter boundary handling.

**Nudged Output:**

```
The loop includes index 0;
  using '>' avoids the extra
  emit ...
```

**Local Criteria (excerpt):**

```
- Boundary correctness for k=0
- Consistency with unit tests
- Minimal patch footprint
```

---

**Local Critic Prompt (Compare Free vs. Nudged)**

Provide a concise verdict comparing the Free and Nudged outputs:
- Does the nudged output better satisfy the criteria?
- If yes, extract the minimal actionable change (patch-ready).
- If no, state the gap in one sentence.

**Output schema:**

```
# Verdict: (Better|Tie|Worse)
# Rationale: <one sentence>
# Action:
- file: src/lib/math.py
  func: top_k
  change: "while i >= k:" -> "while i > k:"
```

The overall task-independent quality score is computed as: $Q_{indep} = \sum_{i=1}^{6} w_i \cdot \frac{r_i - 1}{4} \times 10$, where $r_i \in \{1, \ldots, 5\}$ is the rating for dimension $i$ and weights $w_i$ are: Clarity (0.20), Completeness (0.20), Consistency (0.15), Context (0.15), Reasoning (0.15), Format (0.15).

**Task-dependent performance criteria.** These criteria are initialized generically and refined after the nudged phase based on global performance signals:

- **Functional correctness (0–10):** Does the output achieve the intended task objective?
- **Constraint satisfaction (0–10):** Are all task-specific constraints and requirements met?
- **Quality indicators (0–10):** Task-specific metrics such as test coverage for code generation nodes, relevance scores for retrieval nodes, or accuracy for reasoning nodes.

**Comprehensive evaluation template.** The complete TEP local critic evaluation template integrates both assessment dimensions with clear output specifications:

---

**TEP Local Critic Evaluation Template**

**System Role:** You are a local quality assessor for a compound AI system node.

**Evaluation Framework:**
*Task-Independent Quality Assessment (1–5 scale):*
- **Structural Clarity**: Organization and logical flow
- **Specification Completeness**: Coverage of input requirements
- **Internal Consistency**: Freedom from contradictions
- **Context Integration**: Utilization of parent node outputs
- **Reasoning Transparency**: Clarity of decision rationale
- **Format Compliance**: Adherence to output schema

*Task-Dependent Performance Assessment (1–5 scale):*
- **Functional Correctness**: Achievement of task objective
- **Constraint Satisfaction**: Compliance with task-specific requirements
- **Quality Indicators**: Context-specific metrics (e.g., test coverage, relevance)

**Input Context:**
- *Node Output*: $\{$z_v$\}$
- *Parent Context*: $\{$z_pa(v)$\}$
- *Task Schema*: $\{$schema_v$\}$

**Required JSON Output:**

```
{
  "task_independent": {
    "structural_clarity": <1-5>,
    "completeness": <1-5>,
    "consistency": <1-5>,
    "context_integration": <1-5>,
    "reasoning_transparency": <1-5>,
    "format_compliance": <1-5>
  },
  "task_dependent": {
    "functional_correctness": <1-5>,
    "constraint_satisfaction": <1-5>,
    "quality_indicators": <1-5>
```

```
  },
  "actionable_feedback": "<specific improvement suggestions>",
  "overall_score": <computed weighted average>
}
```

The key innovation is that task-dependent criteria are updated after the nudged phase, incorporating global performance signals without requiring multi-hop backpropagation.

## D.1 TEMPERATURE TUNING STRATEGY

Rather than using fixed temperatures or complex annealing schedules, TEP employs a simple random sampling strategy:

- Sample $\theta_v^{\text{temp}} \sim \mathcal{U}(0.3, 0.9)$ uniformly for each free phase iteration
- Lower temperatures (0.3–0.5) encourage exploitation of current knowledge and refinement of existing solutions
- Higher temperatures (0.6–0.9) promote exploration of alternative approaches and escape from local optima
- No staged approach—continuous random sampling naturally balances exploration and exploitation

This simple strategy avoids the complexity of adaptive temperature schedules while maintaining sufficient diversity in the equilibrium search.

## D.2 EQUILIBRIUM CONVERGENCE CRITERION

A node $v$ is considered to have reached free equilibrium when one of the following conditions is met:

1. **Score stabilization:** The variance of critic scores across $k = 3$ consecutive evaluations falls below threshold $\epsilon = 0.5$
2. **Non-substantive improvements:** Suggested changes become primarily stylistic rather than functional (determined by keyword analysis of feedback)
3. **Iteration budget:** Maximum of 20 refinement iterations reached (practical constraint for efficiency)

**Implementation efficiency.**   The local nature of free phase optimization enables several computational optimizations:

- **Parallel execution:** All nodes can reach equilibrium independently, enabling parallel computation
- **Caching:** Equilibrium states can be cached and reused when parent outputs remain unchanged
- **Early stopping:** Nodes achieving high initial scores (8/10 average) can skip refinement

These implementation details demonstrate that TEP's free phase provides a practical and efficient alternative to global textual backpropagation while maintaining optimization quality through structured local evaluation.

## D.3 TEP CONVERGENCE ANALYSIS

Figure 5 provides detailed convergence analysis of TEP on PubMedQA, demonstrating distinct optimization dynamics at local and global levels. Subplot (a) shows free phase local critic scoring over 20 iterations on a 0-10 scale, where scores progress from initial values around 5.2 toward the equilibrium threshold of 8.0. The characteristic fluctuations demonstrate non-smooth equilibrium-seeking behavior as the local critic iteratively refines prompts, with temporary score decreases reflecting the

Table 3: Solution optimization performance.

| Method | Object Counting w/ Qwen 2.5 (Acc.) | Object Counting w/ Llama 3.2 (Acc.) |
|---|---|---|
| CoT | 60.2 | 72.4 |
| TextGrad | 64.5 | 74.2 |
| TextGrad w/ Sum | 58.6 | 68.9 |
| TEP (Ours) | **67.1** | **81.6** |

exploratory nature of local optimization that can lead to better long-term solutions. Subplot (b) illustrates global convergence across different TEP outer iterations, showing performance improvements from the CoT baseline of 57.3% to the final TEP performance of 62.0%. Notable are the steady fluctuations from outer iteration 25 onwards, demonstrating mature optimization behavior where the system maintains consistent performance variations while achieving incremental gains.

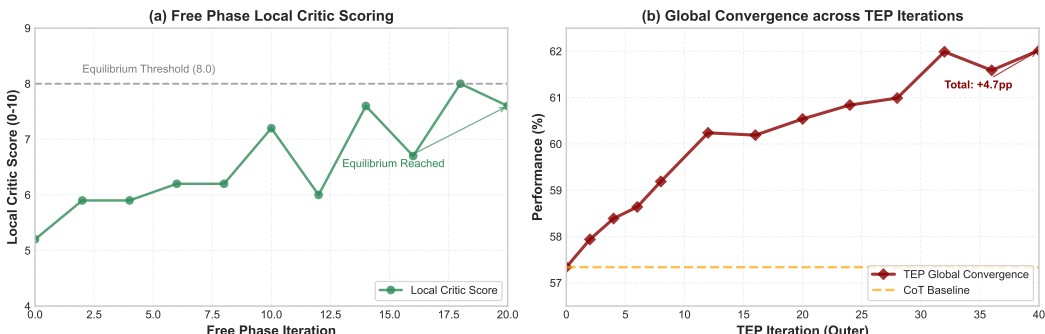

Figure 5: TEP convergence analysis on PubMedQA. (a) Free phase local critic scoring over 20 iterations on 0-10 scale, showing characteristic fluctuations during equilibrium-seeking toward threshold of 8.0. (b) Global convergence across 40 TEP iterations with steady fluctuations from iteration 25 onwards, demonstrating cumulative performance improvements through repeated optimization cycles.

### D.4 EXTENDED EXPERIMENTS: TEP ADVANTAGE ACROSS MODELS

We extend our experiments on solution optimization. To assess whether TEP generalizes across models on the same task, we evaluate our object counting task with the Llama 3.2 11 B model and Qwen 2.5 7B model. As shown in Table 3, TEP consistently outperform other baselines.

### D.5 COMPUTATION COSTS ANALYSIS

A central challenge in measuring computational cost is selecting a metric that is both stable and comparable across settings. While latency can provide context, it fluctuates heavily with region, time, hardware, and provider load. Iteration counts (or API calls) are also uninformative, as different methods may consume vastly different numbers of tokens per iteration. As noted in Sec. 4.3, TextGrad's token usage grows exponentially with workflow depth.

Thus in this paper, we quantify computational cost in terms of token usage rather than wall-clock latency or iteration counts, because it is the most faithful and standardized measure of computational cost in modern LLM systems. Specifically, commercial LLM API providers (e.g., OpenAI[1], Google[2], DeepSeek[3]) charge strictly by token count. Also, deep compound AI system optimization is dominated by token generation and context processing. Moreover, the depth-scaling failure mode we identify, where textual gradients explode or vanish, is tightly linked to the per-iteration token usage (discussed in Sec. 4.3). Taken together, the standard bill method and our specific motivation make token usage a natural, deployment-relevant, and motivation-aligned measure of computational cost in our setting.

---

[1] https://platform.openai.com/docs/pricing
[2] https://ai.google.dev/gemini-api/docs/pricing
[3] https://api-docs.deepseek.com/quick_start/pricing

Table 4: Computational cost (token usage per iteration) for different methods and scales.

| Scale | CoT | TextGrad | TextGrad w/ Sum. | TEP |
|-------|-----|----------|------------------|-----|
| ×1 | 841 | 1856 | 1673 | 1902 |
| ×2 | 1450 | 4224 | 2740 | 2587 |

To better demonstrate the efficiency, we demonstrate the token usage under task PubMedQA and shown in Table 4, TEP has comparable token usage to TextGrad at the smaller scale, but grows much more slowly as the scale increases, leading to substantially less token cost at most ×2 for TextGrad with at scale 2.

## E  SOLUTION OPTIMIZATION DEPTH-SCALING ANALYSIS

This section provides detailed empirical analysis of textual gradient failure modes in solution optimization tasks, complementing the main results presented in Section 4.4.

### E.1  EXPERIMENTAL DESIGN FOR DEPTH-DEPENDENT ANALYSIS

To empirically validate textual gradient pathologies in solution optimization, we conduct a controlled experiment on Object Counting with increasing problem complexity. We design a *nesting depth* parameter $d \in \{1, 2, 3, 4, 5\}$ that controls the number of hierarchical containment levels. At depth $d$, problems require $d$ sequential multiplication steps followed by aggregation and subtraction operations, creating computation graphs with $2d + 2$ nodes.

**Problem Structure.**  For depth $d$, we generate problems following the template: "There are $n_1$ containers. Each container holds $n_2$ sub-containers; each sub-container holds $n_3$ items; ... ; $k$ items are discarded. How many remain?" The nesting creates a chain of $d$ multiplication operations: $n_1 \times n_2 \times \cdots \times n_d$, followed by subtraction. This design ensures that:

- Each intermediate step produces verifiable results
- Errors compound through the calculation chain
- Feedback must propagate through increasingly long solution paths

### E.2  DETAILED RESULTS: EXPLODING AND VANISHING GRADIENTS

**Exploding Gradients: Exponential Message Growth.**  Figure 6(a) demonstrates that TextGrad's feedback messages exhibit exponential growth with depth. Token count increases from 3K at $d = 1$ to over 40K at $d = 5$—a 13× increase that exceeds most LLM context windows. This explosion occurs because each intermediate calculation requires detailed feedback about numerical errors, unit conversions, and logical dependencies, which compounds multiplicatively through the solution chain. The growth follows $B(d) \approx 3000 \times 2.1^d$, confirming our theoretical prediction of exponential scaling.

**Vanishing Gradients: Information Decay Under Compression.**  When TextGrad employs summarization to manage token budgets, gradient specificity degrades exponentially with depth. Figure 6(b) shows TextGrad with summarization declining from 84% at $d = 1$ to merely 28% at $d = 5$. Compressed feedback loses actionable details—transforming specific corrections like "In step 3, multiply 12 boxes by 40 screws per box to get 480, not 120" into generic suggestions like "Check arithmetic in multi-step calculations." This validates our theoretical analysis of information decay under bounded token constraints.

**TEP's Graceful Degradation.**  TEP exhibits fundamentally different scaling behavior, maintaining near-constant token complexity and showing only modest performance degradation (81% to 79% accuracy from $d = 1$ to $d = 5$). By eliminating multi-hop gradient chains in favor of local equilibrium computations, TEP avoids the exponential failure modes that plague global optimization methods while achieving consistent optimization effectiveness across all depths.

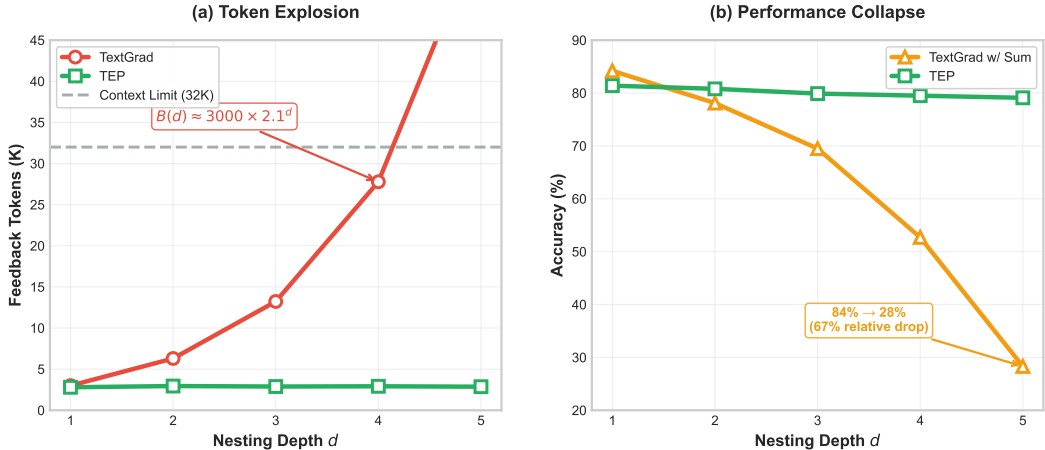

Figure 6: Detailed analysis of textual gradient failure modes in solution optimization. (a) **Token explosion**: TextGrad feedback grows exponentially with depth ($3K \rightarrow 40K$ tokens). (b) **Performance collapse**: TextGrad with summarization shows severe degradation, while TEP maintains robust performance.

**Implications for Solution Optimization.** These results confirm that solution optimization faces even more severe gradient pathologies than prompt optimization. While prompt feedback can be somewhat generic ("make this clearer"), solution feedback must be mathematically precise to enable correction of specific computational errors. The exponential degradation in both token count and specificity makes global textual backpropagation impractical for complex reasoning chains, validating TEP's local approach for solution optimization tasks.

## F LARGE LANGUAGE MODEL USAGE STATEMENT

Large Language Models were used to aid and polish the writing of this paper, including improving clarity, grammar, and presentation of technical content. LLMs assisted with refining explanations, enhancing readability, and ensuring consistent terminology throughout the manuscript.

