# OpenReview forum: "Textual Equilibrium Propagation for Deep Compound AI Systems"
_ICLR.cc/2026/Conference — ICLR 2026 Poster_

### Official Review · Reviewer_xN4N · 2025-10-31

**Soundness:** 3
**Presentation:** 3
**Contribution:** 3
**Rating:** 6
**Confidence:** 2

**Summary:**

The work (i) formalizes exploding and vanishing textual gradients in global textual backprop frameworks like TextGrad, (ii) introduces a depth-robust, node-local alternative inspired by Equilibrium Propagation, and (iii) reports consistent empirical gains on multi-step QA, retrieval, and code-generation suites, alongside ablations and depth-scaling analyses.

**Strengths:**

1. This paper clearly identifies and formalizes depth-dependent pathologies (exploding/vanishing textual gradients) with simple but compelling measures for message length and specificity.

2. This paper presents a principled, modular algorithm (free-phase local refinement; nudged-phase bounded edits) with parallelizable node-local procedures and clean pseudocode.

3. This paper provides breadth across tasks (PubMedQA, STARK-PRIME, HotpotQA, BigCodeBench) and shows consistent improvements, with larger gains at greater depth.

**Weaknesses:**

1. The novelty is unclear because the “free phase + nudged edits” design is very close to existing self-refinement and actor–critic prompt editing methods, and the paper does not include matched-budget comparisons against those strongest local baselines.

2. The definition of “equilibrium” relies on a heuristic stopping rule (score stabilization over three evaluations or a cap of 20 iterations), and the paper does not provide a sensitivity analysis to show that the results are not artifacts of this choice.

3. The computational cost of TEP is underreported; running 20 free iterations plus 40 nudged iterations per node appears expensive, and the paper does not normalize token, latency, or dollar costs against the baselines.

**Questions:**

1. Can you precisely distinguish TEP from prior local methods such as Self-Refine, PACE (actor–critic editing), and verifier-guided prompt tuning, beyond the free/nudged phrasing?

2. Which concrete properties of TEP (and not general self-refinement) are necessary for the reported gains?

---

> ### Author Response · Authors · 2025-11-24
>
> We thank the reviewer for **acknowledging that our identified exploding and vanishing textual gradient is compelling, our method is principled and modular, and our experiments cover broad range of tasks and show consistent improvement**. Below we address your comments point by point.
>
> ---
>
> **W1 & Q1. Novelty: Distinction from self-refinement and actor-critic prompt editing methods.**
> We appreciate the opportunity to clarify TEP's novelty. While TEP shares surface similarities with self-refinement (iterative improvement) and actor-critic editing (critic-guided updates), it is fundamentally different in 1) **problem scope**, 2) **motivation**, and 3) **algorithmic mechanism**.
>
> **Problem Scope**. Methods like Self-Refine and PACE focus on **single-prompt optimization**: given one prompt and one task, they iteratively refine that prompt or its output using self-feedback or an actor–critic loop. These methods operate at the **prompt level and do not explicitly model a multi-node computation graph or cross-agent dependencies in multi-agentic workflow**. **TextGrad** generalizes this idea and provides an **autograd-style engine by modeling a computation graph**. However, its experiments primarily optimize relatively shallow systems. In contrast, **TEP is specifically designed for deep compound AI systems with  multiple prompts to optimize**. It introduces depth-scaling issues (i.e., exploding and vanishing textual gradient identified in our paper) that single-prompt refinement methods (Self-Refine and PACE) neither model nor coordinate.
>
> **Motivation**. Self-Refine and PACE are motivated by **improving prompt quality and reducing human effort for editing prompts** in a single-prompt setting. In contrast, **our work is motivated by depth-scaling issues in deep compound AI systems**. As acknowledged by **reviewers xN4N, ccip and DC6B**, a **core contribution of our paper is to identify and analyze exploding and vanishing textual gradients** as new failure modes of **textual optimization in this scaling setting**. To our knowledge, we are the first to study these depth-scaling phenomena systematically. Standard self-refinement methods do not encounter these issues by design, since they do not model multi-node computation graphs with significant depth, and therefore they do not provide mechanisms to address them.
>
> **Algorithmic Solution**. Rather than relying on **simple iterative editing loops**, TEP fundamentally differs from standard actor–critic or self-refinement methods by employing a unique **two-phase** dynamic (Free phase: acting as a relaxation process where agents settle into a stable, self-consistent equilibrium state, and Nudge phase: using a forward-signaling mechanism via proximal prompt edits). This two-phase approach allows TEP to align agents with global goals without the signal degradation and context explosion in deep compound AI systems.
>
> To address your concern, we have updated **Section 2** to further contrast TEP with these baselines, and clarified that TEP is orthogonal to these single-prompt optimization methods rather than a minor variant.
>
> ---
>
> **W2. Technical Detail: hyperparameter selection.**
> Thank you for your question. **We respectfully disagree that our results are artifacts of the stopping rule.** In **Appendix D.3, Figure 5(a)**, we provide an empirical study of the equilibrium trend during the free phase, showing that the score consistently stabilizes well before reaching the iteration cap. Based on this observed convergence behavior, we set the maximum number of free-phase iterations to 20.

---

> ### Author Response · Authors · 2025-11-24
>
> ---
>
> **W3. Computation cost.**
> In our setting, we measure computational cost by token usage. When optimizing deep compound AI systems with LLM APIs, the dominant cost is token usage. This matches real-world deployments, as commercial providers (e.g., OpenAI, Google, DeepSeek) bill based on tokens. (New): While latency can provide context, it fluctuates heavily with region, time, hardware, and provider load. Iteration counts (or API calls) are also uninformative, as different methods may consume vastly different numbers of tokens per iteration. As noted in **Sec. 4.3**, TextGrad’s token usage grows exponentially with workflow depth.
>
> **Existing computational cost analysis in Sec. 4.3**. In **Figure 3 (a)**, we provide a computational cost comparison between TextGrad and our TEP. There we clearly show that TextGrad leads to rapidly growing feedback context length and increasing token usage, while TEP keeps in a much steady region of token usage.
>
> **New detailed experiments and analysis**. To further strengthen the computational cost analysis, we add a new table on reporting token usage for different baselines at two workflow scales in the PubMedQA task. Full details and description in **Appendix D.5**.
>
> | Scale | CoT | TextGrad | TextGrad w/ Sum. | TEP |
> |-------|-----|----------|------------------|-----|
> | ×1    | 841 | 1856     | 1673             | 1902 |
> | ×2    | 1450| 4224     | 2740             | 2587 |
>
> As shown in the above table, TEP has a close cost to TextGrad at $\times 1$. But TEP token usage grows much more slowly with scale, and it is substantially lower cost at $\times 2$ compared to TextGrad.
>
> ---
>
> **Q2. Which properties of TEP make the gains?**
> As described in **Sec. 3.3 and ablation studies in Sec. 4.5**, the gain comes from three part:
> - **Local equilibrium with rubric-based critics (free phase)**. Each node is first driven to a local equilibrium under a rubric-based critic, which stabilizes its behavior before global nudging. This reduces noisy, unstable updates and mitigates exploding/vanishing textual gradients along the chain.
> - **Slight, global-guided edits (nudged phase)**. In the nudged phase, we use small, proximal prompt edits rather than aggressive rewrites. This leads to more reliable improvements and avoids destabilizing downstream agents.
> - **Interaction between local and global updates**. Alternating between local equilibria and global nudging yields better performance than applying either local refinement or global nudging alone as shown in our ablation studies in **Sec. 4.5**.

---

### Official Review · Reviewer_HWwQ · 2025-11-01

**Soundness:** 4
**Presentation:** 4
**Contribution:** 3
**Rating:** 8
**Confidence:** 4

**Summary:**

This paper addresses the issue of vanishing and exploding textual gradients in compound AI systems by introducing Textual Equilibrium Propagation (TEP). This framework, inspired by equilibrium propagation in energy-based models employs two phases: iterative refinement and nudging. Experimental results indicate that the proposed framework achieves superior performance over TextGrad and offers a robust approach for black-box optimization.

**Strengths:**

- The paper is clearly written, and the inclusion of illustrative examples enhances clarity and comprehension.
- Extensive experiments are conducted, demonstrating consistent empirical gains.
- Evaluation includes both prompt optimization and solution optimization tasks, even though the latter is limited to two datasets.

**Weaknesses:**

- The rationale behind the choice of large language models (LLMs) is insufficiently explained.
- There is limited discussion of potential LLM biases and how they may influence results.
- Details regarding hyperparameter selection and tuning are lacking.
- The paper does not analyze the trade-off between performance gains and the associated increase in computational cost.

**Questions:**

- What is the rationale for selecting the specific large language models (LLMs) used for different tasks?
- How does the size of the LLM influence the performance of various components within the proposed framework?

---

> ### Author Response · Authors · 2025-11-24
>
> We thank the reviewer for **recognizing that our method is well-motivated, presentation is clear and our experiments are comprehensive**. Below we address your comments point by point.
>
> ---
>
> **W1 & Q1. Clarification LLM choice and model size.**
> Thank you for your question. Our choice of LLMs is driven by **task difficulty and domain**. For complex prompt-optimization tasks that require long-horizon reasoning and reliable self-critique, we use frontier models such as GPT-4o and Claude (the latter is particularly strong on coding and complex retrieval). For simpler solution-optimization tasks, we use lighter models such as Llama 3, which already achieve reasonable base performance and are more cost-efficient.
>
> ---
>
> **Q2. LLM model size influence.**
> Regarding model size, **TEP shares the same self-improvement paradigm as TextGrad**: the base LLM must be capable of meaningful self-correction on the target task. Very small or lacking target domain knowledge models often fail to generate or evaluate high-quality refinements, making any textual optimization framework ineffective (as also discussed in the **TextGrad paper Sec. 3.4 Final Para**). We clarify the LLM choice and model size impact in the revised manuscript (see **Appendix A**).
>
> To further answer your question, we further provided experiments with within the same task using different size of models. Specifically, we evaluate the same object-counting task using two different LLM: Qwen 2.5 (7B) and Llama 3.2 (11B). More details are provided in **Appendix D.4**.
>
> | Method        | Qwen 2.5 (Acc.) | Llama 3.2 (Acc.) |
> |--------------|-------------------------------------|--------------------------------------|
> | CoT          | 60.2                                | 72.4                                 |
> | TextGrad     | 64.5                                | 74.2                                 |
> | TextGrad w/ Sum | 58.6                             | 68.9                                 |
> | TEP (Ours)   | **67.1**                            | **81.6**                             |
>
> As shown in the above table, TEP consistently outperforms all baselines, achieving improvements of 3% on Qwen and 7% on Llama. **TEP tends to have more significant improvement with larger LLMs.**
>
> ---
>
> **W2. Discussion: LLM bias.**
> We thank the reviewer for highlighting the issue. We fully acknowledge that the bias associated with "LLM-as-a-judge" bias, like any framework relying on model-generated feedback, TEP is subject to the inherent biases of the critic LLM. We initially touched upon this in **Sec. 1 (Introduction, Para 3)** and **Sec. 2 (Related Work, Para 2)**, noting that global backpropagation methods (like TextGrad) risk amplifying these evaluation biases as they propagate through deep chains.
>
> In deep compound AI systems, **bias can accumulate**. A small judgement error in a downstream node can compound as it is backpropagated upstream. **TEP addresses this specifically through the Free Phase and Nudged Phase.** In the free phase, iterative local updates under a rubric-based critic stabilize each node. In the nudged phase, proximal prompt edits are guided directly by the final task objective (i.e. final ground truth labels). Together, these steps reduce the systematic bias (as shown in **Sec. 4.3**) that accumulates when feedback is propagated step by step along the chain. Following your valuable suggestion, we have expanded our discussion on bias in **Appendix A.4**.
>
> ---
>
> **W3. Technical Detail: hyperparameter.**
> To keep the comparison fair, we align all other hyper-parameters across baseline methods, including initial prompts, temperature setting, node LLM model and workflow choices, so that the observed performance and computational cost difference is driven primarily by the baseline difference. We also included more detailed hyperparameters, configurations and computational cost in **Appendix C and D**.

---

> ### Author Response · Authors · 2025-11-24
>
> ---
>
> **W4. Computational cost.**
> Thank you for your question. We would like to clarify that we do measure computational cost using token usage, and TEP proves effective with noticeably lower token cost. Following your suggestion, we have also included additional analysis to further support this point.
>
> In our setting, we measure computational cost by token usage. When optimizing deep compound AI systems with LLM APIs, the dominant cost is token usage. This matches real-world deployments, as commercial providers (e.g., OpenAI, Google, DeepSeek) bill based on tokens.
>
> **computational cost analysis in Sec. 4.3**. In Figure 3 (a), we provide a computational cost comparison between TextGrad and our TEP. There we clearly show that TextGrad leads to rapidly growing feedback context length and increasing token usage, while TEP keeps in a much steady region of token usage.
>
> **Additional experiments and analysis**. To further strengthen the computational cost analysis, we add a new table on reporting token usage for different baselines at two workflow scales in the PubMedQA task. Full details and description in **Appendix D.5**.
>
> | Scale | CoT | TextGrad | TextGrad w/ Sum. | TEP |
> |-------|-----|----------|------------------|-----|
> | ×1    | 841 | 1856     | 1673             | 1902 |
> | ×2    | 1450| 4224     | 2740             | 2587 |
>
> As shown in the above table, TEP has a close cost to TextGrad at $\times 1$. But TEP token usage grows much more slowly with scale, and it is substantially lower cost at $\times 2$ compared to TextGrad.
>
> Overall, these analyses demonstrate that TEP achieves strong performance improvements while maintaining substantially lower token cost.

---

### Official Review · Reviewer_ccip · 2025-11-01

**Soundness:** 2
**Presentation:** 3
**Contribution:** 3
**Rating:** 6
**Confidence:** 3

**Summary:**

The paper tackles a timely and relevant problem: the optimization of deep compound AI systems that perform multi-step reasoning and multi-tool coordination at realistic scales. The authors identify that current textual optimization methods—particularly TextGrad—encounter depth-scaling failures analogous to the vanishing and exploding gradient problems in neural networks. To address this, they propose **Textual Equilibrium Propagation (TEP)**, a two-phase optimization method inspired by equilibrium propagation in energy-based models.
In TEP, global feedback propagation is replaced by local refinement in a “free phase”, where each node independently updates its prompt using a local LLM critic until equilibrium is reached, and a “nudged phase,” where small task-specific perturbations are introduced to align local equilibria with the global objective. While the authors cite conceptual inspiration from equilibrium propagation, the mathematical connection is loose—TEP does not define or optimize an explicit energy function ($E(x, \theta)$).

**Strengths:**

1. The paper clearly articulates the depth-scaling issues of existing methods and communicates them effectively through illustrative figures.
2. The proposed method has well-defined computational properties: local optimization has bounded token complexity ($O$(1)) per node versus exponential (*O*($2^{depth}$)) complexity in multi-hop textual backpropagation, allowing scalability to deeper systems (10–20+ nodes).
3. Empirically, TEP demonstrates strong performance improvements over TextGrad on multiple benchmarks (PubMedQA, STaRK-PRIME, HotpotQA, BigCodeBench).
4. The depth-scaling experiments (Fig. 3) are particularly compelling: TextGrad’s feedback length grows from 2K to 32K tokens with decreasing update effectiveness (36%→5%), while TEP maintains near-constant token complexity and only modest performance degradation (37%→33%).
5. The ablation studies on TEP’s free and nudged phases effectively show that both components are necessary for the observed gains.

**Weaknesses:**

1. The connection to classical Equilibrium Propagation (Scellier & Bengio, 2017) remains metaphorical rather than formal. TEP does not define a differentiable energy function, nor does it derive a learning rule corresponding to a negative gradient step. As a result, claims of “consistent descent direction” and “convergence under standard conditions” are not theoretically substantiated in the paper.
2. The “convergence analysis” in Appendix A is largely qualitative. It assumes information contraction and bounded channel capacity but does not provide a formal proof of convergence or conditions guaranteeing equilibrium stability.
3. Computational cost analysis is incomplete. While token complexity is analyzed, the paper omits wall-clock time or total iteration comparisons against baseline methods. The trade-off between token efficiency and the number of required refinement steps remains underexplored.
4. The experimental section would benefit from deeper analysis of model generality. Although the authors report results using different base LLMs (GPT-4o, Claude 3 Haiku, Llama 3.2), there are no controlled ablations where the same task is optimized using multiple base models. This weakens the generalization claim.
5. It is unclear whether the same model or a separate model serves as the critic LLM for local refinement. Clarifying this choice (and its impact on stability or bias) would help contextualize results.

**Questions:**

1. Could the authors provide more rigorous mathematical support for the claimed “local optimization until equilibrium” behavior? In particular, can they formalize the contraction assumptions or provide sufficient conditions under which the refinement operator is guaranteed to converge?
2. Clarify whether “consistent descent direction” refers to an empirical observation or a theoretical guarantee. If the latter, please include a proof or clearly stated theorem.
3. Include wall-clock comparisons and iteration counts to quantify the trade-off between per-node efficiency and convergence speed.
4. Strengthen generalization claims by running at least one benchmark task with multiple base models under the same conditions.
5. Specify whether the critic LLM is identical to the base model or distinct, and discuss how this choice affects equilibrium stability and sample efficiency.

---

> ### Author Response · Authors · 2025-11-24
>
> We thank the reviewer for **acknowledging that our motivation is clear and well-presented, and our empirical studies are compelling and ablation studies are comprehensive**.  Below we address your comments point by point.
>
> ---
>
> **W1 . Clarification: Connection to classical equilibrium propagation.**
> Thank you for the question. In our setting, the TextGrad-style computation graph is **non-differentiable**: nodes are black-box LLM APIs and signals are discrete text rather than continuous activations. Because of this, TEP, while inspired by classical equilibrium propagation, is not a direct, formal instance of EP on a differentiable computation graph. **This is not a flaw of TEP, but a structural limitation of applying TextGrad-like optimization in deep compound AI systems.**  We clarified this relationship and limitation in **Sec. 1 Paragraph 2 and 3**.
>
> ---
>
> **W2. Clarification: Convergence Analysis.**
> Our convergence analysis in appendix is mainly on qualitative description not a formal proof, as mentioned before, this a **non-differential** computation graph which makes conventional convergence hard to directly apply. Our goal is to describe an **operational notion** of convergence and related concepts, and do empirical studies to verify our hypothetical intuition of the concept textual gradient method convergence, definition of exploding and vanishing textual gradient. To avoid overstating the theoretical guarantees, we have revised the **Appendix A** title to "Analysis" (not formal theoretical analysis) to more accurately reflect its intent. Additionally, our remark on “consistent descent direction” refers to an empirical observation, which is shared by our method and TextGrad rather than being presented as a formal theorem.
>
> ---
>
> **W3. Technical Detail: Computational cost.**
> Thank you for your question. A central challenge in measuring computational cost is selecting a metric that is both stable and comparable across settings. While latency can provide context, it fluctuates heavily with region, time, hardware, and provider load. Iteration counts (or API calls) are also uninformative, as different methods may consume vastly different numbers of tokens per iteration. As noted in **Sec. 4.3 Figure 3 (a)**, TextGrad’s token usage grows exponentially with workflow depth.
>
> Thus in this paper, we quantify computational cost in terms of **token** usage, because it is the most faithful and standardized measure of computational cost in deep compound AI systems. Specifically, commercial LLM API providers (e.g., OpenAI [1], Google [2], DeepSeek[3]) charge strictly by token count. Also, deep compound AI system optimization is dominated by token generation and context processing. Moreover, the depth-scaling failure mode we identify, where textual gradients explode or vanish, is tightly linked to the per-iteration token usage (discussed in **Sec. 4.3**). Taken together, the standard bill method and our specific motivation make token usage a natural, deployment-relevant, and motivation-aligned measure of computational cost in our setting.
>
> **New detailed experiments and analysis**. To further strengthen the computational cost analysis, we add a new table on reporting token usage for different baselines at two workflow scales in the PubMedQA task. Full details and description in Appendix D.5.
>
> | Scale | CoT | TextGrad | TextGrad w/ Sum. | TEP |
> |-------|-----|----------|------------------|-----|
> | ×1    | 841 | 1856     | 1673             | 1902 |
> | ×2    | 1450| 4224     | 2740             | 2587 |
>
> As shown above, TEP’s cost is comparable to TextGrad at the ×1 scale. However, as the workflow becomes deeper (×2), TextGrad’s token usage increases sharply, while TEP grows much more slowly, resulting in substantially lower cost at larger scales. This demonstrates that TEP maintains a more favorable scaling behavior as depth increases.
>
>
> [1] https://platform.openai.com/docs/pricing
> [2] https://ai.google.dev/gemini-api/docs/pricing
> [3] https://api-docs.deepseek.com/quick_start/pricing

---

> ### Author Response · Authors · 2025-11-24
>
> ---
>
> **W4. Experiments: Model generality.**
> Thank you for your question. Our method have shown strong and consistent performance across multiple base models and across multiple datasets. This provides meaningful evidence of TEP model generality.
>
> Following your suggestion, we further strengthened this by adding controlled ablations on the same task using different base models. Specifically, we evaluate the same object-counting task using two different LLM: Qwen 2.5 (7B) and Llama 3.2 (11B). More details are provided in **Appendix D.4**.
>
> | Method        | Qwen 2.5 (Acc.) | Llama 3.2 (Acc.) |
> |--------------|-------------------------------------|--------------------------------------|
> | CoT          | 60.2                                | 72.4                                 |
> | TextGrad     | 64.5                                | 74.2                                 |
> | TextGrad w/ Sum | 58.6                             | 68.9                                 |
> | TEP (Ours)   | **67.1**                            | **81.6**                             |
>
> As shown in the above table, TEP consistently outperforms all baselines, achieving improvements of 3% on Qwen and 7% on Llama, confirming that the method generalizes well across different model families on the exact same task.
>
> ---
>
> **W5. Clarification: Minor setup detail.**
> Thanks for pointing this out. The critic model is indeed identical to the node model used within the deep compound AI system, and we have now made this explicit in revised **Section 3.3**.

---

### Official Review · Reviewer_DC6B · 2025-11-10

**Soundness:** 3
**Presentation:** 3
**Contribution:** 3
**Rating:** 6
**Confidence:** 4

**Summary:**

This paper investigates textural propagation in large language models (LLMs) to enhance compound AI systems. The authors point out that existing textural propagation approaches, such as textgrad, suffer from two key problems: exploding textural gradients and vanishing textural gradients in the middle context. They propose Text Equilibrium Propagation (TEP), a local learning principle in which a local LLM critic iteratively refines prompts until no further performance gain is observed. The method also introduces forward propagation of prompt edits. Experimental results show that TEP achieves larger improvements than previous methods, especially as task depth increases.

**Strengths:**

1.The problem is well motivated.
2.The proposed method is conceptually sound.

**Weaknesses:**

1. It is not entirely clear how the local critic model operates. Does it require ground truth labels for intermediate outputs?
2. The baseline method Revolve[1] is mentioned.
3. While performance gains are significant, the paper does not analyze the tradeoff between accuracy and computational cost,e.g. how does the computational cost look like for each of the baseline and TEP?

[1]Revolve: Optimizing AI Systems by Tracking Response Evolution in Textual Optimization

**Questions:**

1.Can you explain what is the main difference between the forward signaling and backward feedback chains?
2.In Table 1, the improvement of TextGrad over Chain of Thought (CoT) is relatively small. Do you have any insights into why this happens?

---

> ### Author Response · Authors · 2025-11-24
>
> We thank the reviewer for **recognizing our well-motivated and conceptually sound method**. Below we address your comments point by point.
>
> ---
>
> **W1. Technical Detail: Use of intermediate labels.**
> Thank you for your question. As described in **Sec. 3.3** and **Appendix D**, ground-truth labels are only available for the final output level. For the intermediate node prompt optimization, the feedback comes from a local critic model guided by a structured rubric-based prompt. Further details are provided in **Sec. 3.3** (Free Phase) and **Appendix C**. Following your suggestion, we have further clarified this detail in **Sec. 3.3** of our revision.
>
> ---
>
> **W2. Paper Positioning: REVOLVE baseline.**
> We thank the reviewer for pointing out Revolve [1], a TextGrad-based prompt optimization framework. It tracks the evolution of responses across iterations for a **single prompt** and uses historical messages to stabilize prompt optimization.
>
> **First, we focus on different problem settings.**
> Revolve operates in a **single-prompt setting** and improves TextGrad along the temporal axis. It tracks the evolution of response to help optimization stabilize. In contrast, TEP is designed for deep compound AI systems with **multiple-prompt** to optimize. Our method is built around the structural axis: how to improve multi-prompt optimization in a depth-scaling agentic workflow.
>
> **Second, we tackle different failure modes of TextGrad.**
> Revolve primarily **improves prompt optimization stability over iterations**, helping avoid oscillations or loops between poor solutions for a single prompt. Our work instead focuses on **mitigating vanishing and exploding textual gradient** that arise when optimizing a deep compound AI system. This depth-related failure mode arises in the TextGrad-based optimization framework (including Revolve when extended to deep compound AI systems), which makes TEP complementary rather than competing.
>
> In summary, revolve and our work are **orthogonal work** that targets **different problem settings and stem from different motivations**. We have added the discussion with the Resolve baseline discussion in **Sec. 2** to better clarify our paper positioning.
>
> [1] REVOLVE: Optimizing AI Systems by Tracking Response Evolution in Textual Optimization
>
> ---
>
> **W3. Technical Detail: Computational cost.**
> Thank you for your question. We would like to clarify that we do measure computational cost using token usage, and following your suggestion, we also include additional analysis.
>
> **Token as computational cost**. In our setting, computational cost is measured by **token usage**, which is the dominant cost when optimizing deep compound AI systems through LLM APIs. This aligns with real-world deployment practice: major commercial providers (e.g., OpenAI, Google, DeepSeek) charge strictly based on input/output token count, making tokens the most accurate and reproducible cost metric.
>
> **Existing computational cost analysis in Sec. 4.3**. In **Figure 3 (a)**, we provide a computational cost comparison between TextGrad and our TEP. There we clearly show that TextGrad leads to rapidly growing feedback context length and increasing token usage, while TEP keeps in a much steady region of token usage.
>
> **Additional experiments and analysis**. To further strengthen our computational cost evaluation, we added a new table reporting token usage across different baselines at two workflow scales on the PubMedQA task. Full details and description in **Appendix D.5**.
>
> | Scale | CoT | TextGrad | TextGrad w/ Sum. | TEP |
> |-------|-----|----------|------------------|-----|
> | ×1    | 841 | 1856     | 1673             | 1902 |
> | ×2    | 1450| 4224     | 2740             | 2587 |
>
> As shown above, TEP’s cost is comparable to TextGrad at the ×1 scale. However, as the workflow becomes deeper (×2), TextGrad’s token usage increases sharply, while TEP grows much more slowly. This demonstrates that TEP is more cost-effective at greater depth.
>
> ---
>
> **Q1. Forward signaling and backward feedback.**
> Thank you for your question. As described in **Sec. 3.3**, forward signaling in TEP (used in the nudged phase) lets each local module do slight prompt edits, then propagate these edits forward to observe the final output. This is a “local perturb-and-observe” mechanism instead of requiring step-by-step textual gradient transmit and accumulate every intermediate feedback.
>
> ---
>
> **Q2. TextGrad underperforming.**
> Our interpretation of the underperformance of TextGrad is that the textual gradient works well in single-prompt or relatively short workflow (see **Sec. 1 Paragraph 2**) but struggles in the deep compound AI system. We attribute this to the textual gradient becoming not useful due the observed trend of exploding and vanishing textual gradient. In this regime, the gradients no longer give reliable guidance for updating prompts, and TextGrad can fail to improve performance.

---

### Meta-Review · Area_Chair_1Sij · 2025-12-28

**Summary:**

The paper addresses the scalability limitations of global textual backpropagation (e.g., TextGrad) in deep compound AI systems. The authors identify two key failure modes analogous to neural networks: "exploding textual gradients" (exponential growth of feedback message length) and "vanishing textual gradients" (loss of specificity due to compression/context limits). To mitigate these, the authors propose Textual Equilibrium Propagation (TEP), a method inspired by Equilibrium Propagation in energy-based models. TEP utilizes a local, two-phase optimization process: 1) a "Free Phase" where local critics refine prompts to a stable equilibrium, and 2) a "Nudged Phase" where bounded prompt edits align the system with global task objectives via forward signaling. Experiments on PubMedQA, HotpotQA, STaRK-PRIME, and BigCodeBench demonstrate that TEP outperforms TextGrad, particularly in deeper workflows, while maintaining constant per-node token complexity.

**Reviewer Concerns:**

- Theoretical Connection to Equilibrium Propagation (Reviewer ccip): The reviewer noted that the connection to classical Equilibrium Propagation (Scellier & Bengio) is metaphorical rather than mathematically formal (lacking a differentiable energy function).

Response: The authors acknowledged this, clarifying that TEP applies the structural intuition of EqProp to non-differentiable, discrete LLM computation graphs. They renamed the appendix section from "Theoretical Analysis" to "Analysis" to avoid overclaiming. The Area Chair agrees that the empirical adaptation is valuable even without strict mathematical equivalence.

- Novelty and Baselines (Reviewers DC6B, xN4N): Reviewers asked for differentiation from methods like Revolve, Self-Refine, and PACE.

Response: The authors clarified that Revolve focuses on single-prompt temporal stability, whereas TEP targets multi-node structural depth issues. Similarly, Self-Refine/PACE operate on single prompts without the coordination mechanism required for deep computation graphs.

- Computational Cost and Efficiency (Reviewers ccip, HWwQ, xN4N): Reviewers requested better cost analysis (wall-clock time vs. tokens) and trade-offs.

Response: The authors argued that token usage is the standard, reproducible metric for API-based systems (as latency varies by provider load). They provided a new table showing that while TextGrad's token cost grows exponentially with depth, TEP's cost remains linear/stable, making it significantly more efficient for deep workflows.

- Experimental Details (Reviewers DC6B, HWwQ): Questions regarding the need for intermediate ground-truth labels and model generality.

Response: The authors clarified that TEP does not require intermediate ground truths (using rubric-based critics instead). They also added ablation studies with different model sizes (Llama 3.2 11B, Qwen 2.5 7B) to demonstrate the method generalizes beyond the specific frontier models used in the main paper.

**Reviewer Scores:**

I cannot reliably answer this counterfactual question without putting words in reviewers’ mouths. I will not impute score changes beyond what reviewers explicitly stated in the discussion. I instead provide a faithful synthesis of the discussion outcomes and remaining points of disagreement.

---

### Decision · Program_Chairs · 2026-01-26

Accept (Poster)